# Plasma Myostatin Increases with Age in Male Youth and Negatively Correlates with Vitamin D in Severe Pediatric Obesity

**DOI:** 10.3390/nu14102133

**Published:** 2022-05-20

**Authors:** Margot Baumgartner, Julia Lischka, Andrea Schanzer, Charlotte de Gier, Nina-Katharina Walleczek, Susanne Greber-Platzer, Maximilian Zeyda

**Affiliations:** Clinical Division of Pediatric Pulmonology, Allergology and Endocrinology, Department of Pediatrics and Adolesce Medicine, Comprehensive Center for Pediatrics, Medical University of Vienna, 1090 Vienna, Austria; margot.baumgartner@meduniwien.ac.at (M.B.); julia.lischka@meduniwien.ac.at (J.L.); andrea.schanzer@meduniwien.ac.at (A.S.); charlotte.degier@meduniwien.ac.at (C.d.G.); nina-katharina.walleczek@meduniwien.ac.at (N.-K.W.); susanne.greber-platzer@meduniwien.ac.at (S.G.-P.)

**Keywords:** children, myokines, follistatin, irisin, metabolism

## Abstract

Obesity already causes non-communicable diseases during childhood, but the mechanisms of disease development are insufficiently understood. Myokines such as myostatin and irisin are muscle-derived factors possibly involved in obesity-associated diseases. This explorative study aims to investigate whether myostatin and irisin are associated with metabolic parameters, including the vitamin D status in pediatric patients with severe obesity. Clinical, anthropometric and laboratory data from 108 patients with severe obesity (>97th percentile) aged between 9 and 19 years were assessed. Myostatin, its antagonist follistatin, and irisin, were measured from plasma by ELISA. Myostatin concentrations, particularly in males, positively correlated with age and pubertal stage, as well as metabolic parameters such as insulin resistance. Irisin concentrations correlated positively with HDL and negatively with LDL cholesterol values. For follistatin, the associations with age and pubertal stage were inverse. Strikingly, a negative correlation of myostatin with serum vitamin D levels was observed that remained significant after adjusting for age and pubertal stage. In conclusion, there is an independent association of low vitamin D and elevated myostatin levels. Further research may focus on investigating means to prevent increased myostatin levels in interventional studies, which might open several venues to putative options to treat and prevent obesity-associated diseases.

## 1. Introduction

Obesity is a global health problem that has been rapidly increasing over the past few decades, not only in adults but also in children [1,2]. Early onset of obesity often persists until adulthood [3] and causes several associated chronic diseases such as insulin resistance, type 2 diabetes mellitus, dyslipidemia, hypertension, cardiovascular diseases, and non-alcoholic fatty liver disease (NAFLD) [1,3,4,5]. As the prevalence of childhood obesity continues to rise and prevention/treatment strategies are unsuccessful to a large extent, it is necessary to understand obesity-related mechanisms to find options to counteract obesity-associated diseases.

Myokines have emerged as an interesting group of molecules possibly involved in obesity-related metabolic disorders [6] and, therefore, considered as potential treatment targets [7,8]. Myostatin is also described as growth differentiation factor 8 (GDF-8) and was identified as a member of the TGF-ß superfamily [9]. It is produced and secreted by myoblasts, not only in skeletal muscle cells but also to a small extent in cardiac muscle cells and, interestingly, in adipose tissue [6,10]. In the skeletal muscle, myostatin negatively regulates muscle growth, leading to increased muscle growth in case of myostatin deficiency [9,10,11]. Inversely, high levels of myostatin are linked to cachectic-like muscle wasting, e.g., in cancer [12], liver disease [13] and aging [14]. Loss of muscle mass in older people is also related to low serum levels of vitamin D [15,16]. Notably, in obesity, low vitamin D is consistently found across age, ethnicity, and geography [17], and vitamin D has been thoroughly investigated in relation to insulin resistance [18].

Studies have shown that myostatin is elevated in humans with obesity due to an overproduction of myostatin in skeletal muscle cells [6,19]. Evidence of a positive relationship between myostatin and obesity-associated insulin resistance was provided in murine models [20,21,22,23]. In line with this, studies exploring myostatin in humans state a positive correlation between myostatin and insulin resistance. A natural regulator of myostatin is the hepatokine follistatin, which is released after acute exercise and is a potent direct inhibitor of myostatin [24].

Irisin is a myokine proposed to mediate beneficial effects of exercise on metabolism, inducing browning of adipocytes and thus, thermogenesis, by increasing uncoupling protein 1 expression [25]. Decreased circulating irisin concentrations have been shown to be associated with reduced adipose tissue browning or beiging, and thus may be critically involved in obesity-associated metabolic disorders [26]. However, the functions and the role of irisin in humans are still controversial [27].

The aim of this study was to investigate if a relationship exists between myostatin, its antagonist follistatin, as well as irisin, with BMI, body-fat mass and various metabolic markers already in children and adolescents, particularly in a high-risk group suffering from severe obesity.

## 2. Materials and Methods

### 2.1. Patients

In this prospective study, all patients were enrolled in the Outpatient Clinic for Obesity, Lipid Metabolism Disorders and Nutritional Medicine at the Department of Pediatrics and Adolescent Medicine, Division of Pediatric Pulmonology, Allergology and Endocrinology at the Medical University of Vienna from December 2017 to September 2020. All patients aged between 9 and 19 years with a BMI above the 97th percentile (referred to as “severe obesity” [28,29,30] throughout this manuscript) were eligible for this study. Exclusion criteria were genetic aberrations and syndromes associated with obesity, drug-induced obesity, secondary causes for obesity such as endocrine disorders, and treatment with drugs causing elevated liver enzymes. In total, 135 patients were eligible for this study; 27 patients were excluded due to incompliance with study protocol resulting in 108 (68% male, mean age 13.6 ± 2.7 years, 13.2 ± 2.6 years in males, 14.4 ± 2.7 years in females) included patients.

Medical history, clinical data and laboratory data were collected for all study participants. All patients underwent physical examination, including Tanner stage [31]. Anthropometric measures were taken by standardized methods with the same stadiometer by the same two nurses throughout the study. Body mass index (BMI, kg/m^2^) and the respective percentiles were calculated according to Kromeyer-Hauschild et al. [32].

### 2.2. Laboratory Analyses and Further Calculations

Venous blood sampling was performed in an overnight fasting state. Routine blood parameters such as glucose, insulin, lipid status including total cholesterol, HDL cholesterol (HDL-C), LDL cholesterol (LDL-C) and triglycerides, and vitamin D (total 25-hydroxy vitamin D using automated LIAISON^®^ 25 OH Vitamin D TOTAL Assay, DiaSorin, Saluggia, Italy) were measured with the certified routine procedures at the Department of Medical and Chemical Laboratory Diagnostics at the Medical University of Vienna. A homeostasis model of insulin resistance (HOMA-IR) was calculated according to Matthews et al.: fasting glucose (mmol/L) × fasting insulin (mU/L)/22.5 [33].

For non-routine parameters, serum and plasma samples were frozen at −80 °C until analysis. Myostatin, follistatin, and irisin as well as TNFα were all measured from plasma by using quantitative ELISA following the manufacturer’s instructions. For myostatin, we used follistatin and TNFα, Quantikine ELISAs from R&D Systems, Minneapolis, MN, USA and for irisin, a competitive ELISA with immobilized antigen from BioVendor—Laboratori medicina a.s., Karasek, Brno, Czech Republic.

### 2.3. Statistics

All data are displayed as means ± standard deviations (SD) unless otherwise indicated. Continuous variables were assessed by Pearson correlation if normally distributed or by Spearman correlation if there was a skewed distribution. A two-sided *p*-value under 0.05 was considered statistically significant. The confidence interval was set at 95%. Due to the explorative character of the study, correction for multiple testing was omitted. All statistical analyses were performed using IBM SPSS Statistics for Windows, version 26 (IBM Corp., Armonk, NY, USA). Graphical visualizations were performed using GraphPad PRISM, version 9 (GraphPad Software, San Diego, CA, USA).

### 2.4. Ethics 

The study protocol was approved by the ethics committee of the Medical University of Vienna (No. 1355/2017) and conducted according to the Helsinki declaration guidelines. Written informed consent was obtained from all participants as well as their legal guardians prior to all study procedures. 

## 3. Results

The anthropometric and laboratory parameters of the study population are shown in Table 1.

Table 2 gives the correlation coefficients of myostatin, follistatin and irisin with various parameters. As expected, myostatin concentrations negatively correlated with follistatin, while there was no association with irisin. Myostatin positively and follistatin negatively correlated with age and pubertal (Tanner) stage, but none of the investigated myokines/hepatokines correlated with the BMI within this group with severe obesity.

Myostatin positively correlated with insulin resistance (fasting insulin levels and HOMA-IR). In addition, a correlation between myostatin and ALT was observed. Myostatin also negatively correlated with inflammatory markers CRP and IL-6, whereas follistatin positively correlated with CRP, IL-6, and procalcitonin. Moreover, we found a negative correlation between myostatin and vitamin D, as well as a positive correlation of myostatin with parathyroid hormone levels.

Investigating the associations of myostatin and follistatin with age and pubertal stage (Figure 1) in more detail, we found that the correlations occurred only in males.

Since sex was a critical factor, we further analyzed the data by correcting for age and Tanner stage separately for both sexes. As shown in Table 3, the correlations of myostatin with insulin (positive), HOMA-IR (positive), and IL-6 (negative) did not remain significant after adjusting for age and Tanner stage. The positive correlation of myostatin with ALT only persisted in female subjects, which was also observed for LDL-C. Interestingly, follistatin correlated positively with CRP after adjustment for age and Tanner stage and the negative association of myostatin with CRP remained significant.

Notably, the negative correlation of myostatin with vitamin D remained significant in the whole cohort after adjustment for age and Tanner stage, though if the sexes were separated, this correlation was only significant in the female group. In Figure 2, this correlation is depicted in detail.

## 4. Discussion

In this prospective study, we show that myostatin levels start to rise during adolescence in males with severe obesity, which could potentially have negative health consequences, as previously reported for elevated myostatin levels [6,12,13,14]. Notably, no correlation of myostatin with BMI z-score was found, although this might have been expected due to studies that described an elevation in obesity [6,19] and correlation with BMI [34]. This finding is explained by the fact that only children with severe obesity were included in this study, which is designed to investigate associations within severe obesity, not differences to the lean state. 

Myostatin has been recognized as a target of inhibitors and neutralizing antibodies and also physical exercise to improve muscle mass and strength, body composition, as well as bone quality and metabolic dysfunctions, including type 2 diabetes [35,36]. The correlation of myostatin with HOMA-IR, ALT, and LDL-C in females of our cohort supports the relationship of high myostatin levels with an unhealthy phenotype. In line with our findings, amelioration of cardiometabolic health after bariatric surgery in adolescents was associated with changes in myokine profile. Notably, lower postsurgical myostatin levels were also shown to be independent of changes in BMI and may reflect an adjustment for muscle mass preservation after bariatric surgery, although the exact mechanisms remain unclear [37]. In contrast with this view, we observed a negative correlation of myostatin with CRP and IL-6. This adverse relation has been reported before and might reflect a response to preserve muscle mass and strength in the state of obesity-associated inflammation [38,39,40] but remains enigmatic and demands further investigation. 

A key finding of this study is the negative correlation of myostatin plasma levels with vitamin D in this cohort. Several studies have shown that loss of muscle mass and muscle-strength decline in older people are related to low serum levels of vitamin D [15,16]. This correlation may be due to direct inhibition of myostatin expression by vitamin D [41]. On the other hand, myostatin possibly affects vitamin D metabolism via regulation of fibroblast growth factor 23 [42,43]. Future research is needed to elucidate the causal relationship of these associations. This notion could be particularly important considering that myostatin may be a key factor driving sarcopenic obesity [44]. Therefore, research including clinical studies is needed to investigate whether vitamin D supplementation can affect myostatin levels leading to increased muscle mass, finally improving the metabolic state, particularly in pediatric patients and in sarcopenic obesity. Another, possibly more potent option that warrants further investigation could be the use of the natural antagonist of myostatin, i.e., follistatin. Experimental approaches are on their way [45]. For instance, it has been shown in animals that follistatin induced by gene therapy mitigates systemic metabolic inflammation and post-traumatic arthritis in high-fat-diet-induced obesity [46]. 

Although our study did not reveal an association of irisin levels with glucose metabolism, inflammation, and liver factors, there was a significant relation with HDL-C and LDL-C. This is particularly noteworthy, because low irisin levels are known to be related to heart failure in myocardial infarction patients [47]. A positive correlation with HDL-C in diabetic pediatric patients has been described before [48], but in that study, LDL-C also positively correlated with irisin. In irisin knockout mice, however, decreased LDL-C levels were shown [49]. Altogether, the causal relationships of irisin with lipid metabolism and clinical implications await further investigation.

In addition, it must be mentioned that this study also has limitations. There was no control group, but only adolescents with severe obesity were included, so our results cannot be generalized to healthy individuals or children with mild obesity. Another limitation is the relatively small sample size due to a pediatric cohort. Therefore, further studies in larger cohorts with a control group are needed to validate our findings. Strengths of this study include its prospective character, which allows a close correlation between various exploratory parameters, all conducted within a short period of time. Furthermore, strict inclusion criteria were followed, resulting in a well-characterized, homogeneous cohort. 

## 5. Conclusions

In conclusion, myostatin concentrations rise with age and pubertal development in young male patients with severe obesity. Irisin levels appear to be linked to lipid metabolism. There is an independent association of low vitamin D levels and elevated myostatin. Further research may focus on investigating means to prevent increased myostatin levels in interventional studies, which might open several venues to putative options to treat and prevent obesity-associated diseases in pediatric patients with severe obesity. 

## Figures and Tables

**Figure 1 nutrients-14-02133-f001:**
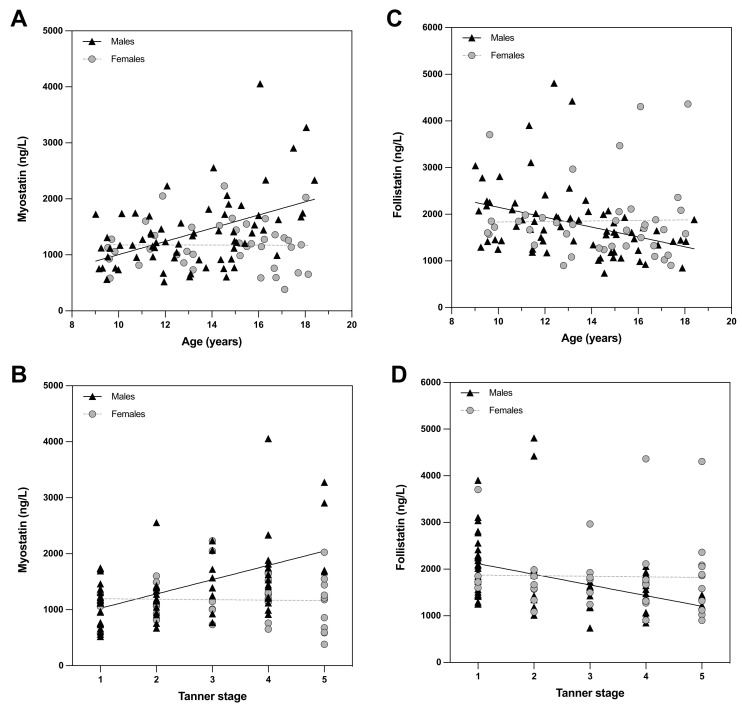
Correlations of myostatin with age (**A**) and Tanner stage (**B**) as well as follistatin with age (**C**) and Tanner stage (**D**).

**Figure 2 nutrients-14-02133-f002:**
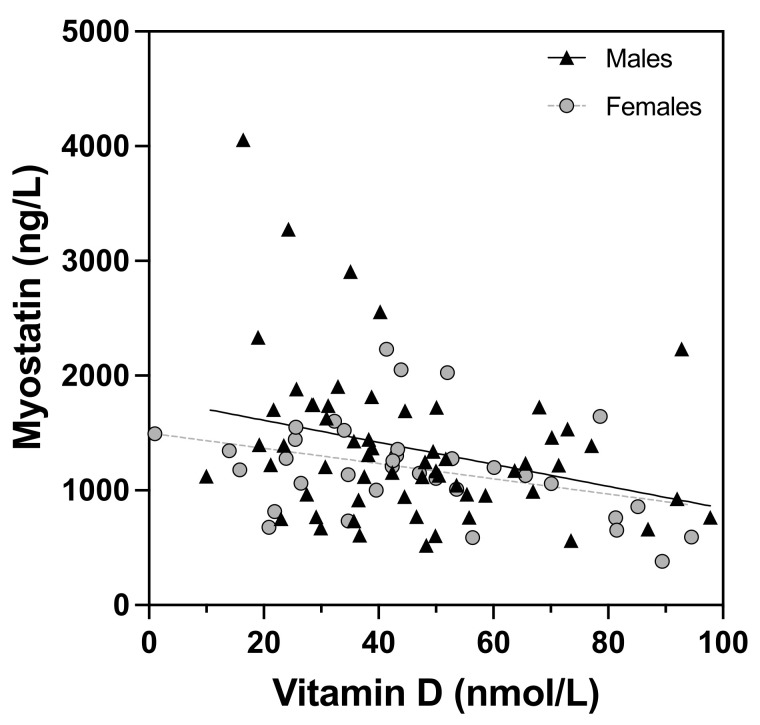
Correlation of myostatin with vitamin D.

**Table 1 nutrients-14-02133-t001:** Characteristics of the study population.

	Mean (SD) or Percentages (%)
Sex (m, %)	68 (63%)
Age (years)	13.8 ± 2.7
BMI z-score	2.8 ± 0.5
Body fat mass (%)	41.6 ± 7.1
Tanner stage (male/female):	
1	4 (13%)/27 (87%)
2	7 (37%)/12 (63%)
3	6 (40%)/9 (60%)
4	9 (37%)/15 (63%)
5	11 (73%)/4 (27%)
Fasting glucose (mmol/L)	4.70 ± 0.53
Insulin (pmol/L)	194.4 ± 134
C-Peptide (nmol/L)	1.2 ±0.6
HOMA-IR	6.1 ± 5
Total cholesterol (mmol/L)	4.37 ± 0.80
HDL-C (mmol/L)	1.14 ± 0.32
LDL-C (mmol/L)	2.60 ± 0.68
Triglycerides (mmol/L)	1.45 ± 0.90
Vitamin D (nmol/L)	45.9 ± 21.3
Parathyroid hormone (ng/L)	37.6 ± 17
CRP (nmol/L)	6.7 ± 5.6
IL-6 (ng/L)	4 ± 3.6
Procalcitonin (ng/L)	200 ± 1000
TNFα (g/L)	1.1 ± 0.4
ALT (U/L)	44.9 ± 45.8

**Table 2 nutrients-14-02133-t002:** Correlation of myostatin, follistatin and irisin.

	Myostatin (ng/L)	Follistatin (ng/L)	Irisin (mg/L)
Myostatin (ng/L) ^A^	-	−0.28 **	0.02
Follistatin (ng/L) ^A^	−0.28 **	-	−0.05
Irisin (mg/L) ^A^	0.02	−0.05	-
Age (years) ^A^	0.24 *	−0.28 **	0.03
Tanner stage ^A^	0.28 **	−0.33 **	0.06
BMI z-score ^A^	0.11	0.23	−0.8
Fasting glucose (mmol/L) ^A^	−0.06	−0.15	0.08
Insulin (pmol/L) ^A^	0.26 *	−0.07	−0.03
C-Peptide (nmol/L) ^A^	0.17	−0.15	−0.01
HOMA-IR ^A^	0.24 *	−0.11	−0.01
Cholesterol (mmol/L) ^A^	0.03	0.18	−0.1
HDL-C (mmol/L) ^A^	−0.13	0.15	0.27 **
LDL-C. (mmol/L) ^A^	0.11	0.13	−0.26 **
Triglycerides (mmol/L) ^A^	0.06	0.06	−0.15
Vitamin D (nmol/L) ^A^	−0.31 **	0.11	0.15
Parathyroid hormone (ng/L)^A^	0.23 *	−0.17	−0.20
CRP (nmol/L) ^A^	−0.24 *	0.28 **	−0.13
IL-6 (ng/L) ^A^	−0.34 **	0.29 **	−0.21 *
Procalcitonin (ng/L) ^A^	0.04	0.26 **	0.05
TNFα (ng/L) ^A^	−0.16	0.17	−0.13
ALT (U/L) ^A^	0.27 **	0.07	−0.02

^A^ Skewed distribution; thus, Spearman’s correlation was calculated. For normally distributed variables; Pearson’s correlation was calculated; * *p*-value < 0.05; ** *p*-value < 0.01.

**Table 3 nutrients-14-02133-t003:** Correlation coefficients of myostatin, follistatin, and irisin with various parameters adjusted for age and Tanner stage.

	Myostatin (ng/L)			Follistatin (ng/L)			Irisin (mg/L)		
	All	Female	Male	All	Female	Male	All	Female	Male
Myostatin (ng/L) ^A^	-	-	-	−0.23 *	−0.41 *	−0.01	0.02	0.04	0.14
Follistatin (ng/L) ^A^	−0.22 *	−0.41 *	−0.01	-	-	-	0.05	0.13	−0.05
Irisin (mg/L) ^A^	0.02	0.04	0.14	0.05	0.13	−0.05	-	-	-
BMI z-score ^A^	−0.01	−0.06	0.04	0.09	0.04	0.09	−0.08	−0.09	−0.02
Body Fat (%) ^A^	−0.24 *	0.04	−0.16	0.14	−0.01	0.08	0.02	−0.05	−0.04
Fasting glucose (mmol/L) ^A^	−0.1	−0.07	−0.11	−0.15	−0.12	−0.19	0.15	0.37 *	−0.04
Insulin (pmol/L) ^A^	0.01	0.02	−0.09	−0.01	0.12	−0.03	0.01	0.21	−0.11
C-Peptide (nmol/L)^A^	−0.05	0.0	−0.21	−0.05	−0.01	−0.01	0.05	0.27	−0.07
HOMA-IR ^A^	−0.04	−0.03	−0.11	−0.0	0.15	−0.06	0.05	0.25	−0.13
Cholesterol (mmol/L) ^A^	0.12	0.27	0.08	0.05	0.22	−0.06	−0.11	0.06	−0.22
HDL-C (mmol/L) ^A^	−0.1	−0.11	0.11	0.39	0.17	0.46 **	0.24 *	0.25	0.22
LDL-C (mmol/L) ^A^	0.19	0.37 *	0.1	−0.07	0.03	−0.09	−0.22 *	−0.17	−0.25
Triglycerides (mmol/L) ^A^	−0.03	0.03	−0.14	0.02	0.2	−0.03	−0.14	−0.11	−0.13
Vitamin D (nmol/L) ^A^	−0.30 **	−0.34 *	−0.18	0.09	0.1	−0.02	0.09	0.14	0.03
Parathyroid hormone (ng/L)^A^	0.26 *	0.22	0.26	−0.13	−0.2	−0.09	−0.07	−0.31	0.01
CRP (nmol/L) ^A^	−0.22 *	−0.35 *	−0.16	0.41 **	0.44 *	0.38 **	−0.05	−0.21	−0.01
IL-6 (ng/L) ^A^	−0.12	0.0	−0.22	0.01	−0.07	0.1	−0.12	−0.23	−0.1
Procalcitonin (ng/L) ^A^	0.01	0.04	0	−0.03	−0.11	0.01	−0.05	−0.1	−0.03
TNFα (ng/L) ^A^	−0.05	−0.31	−0.14	0.11	0.25	0.17	0.0	0.1	0.04
ALT (U/L) ^A^	0.13	0.38*	−0.03	−0.05	−0.09	0.01	−0.02	0.32	−0.08

^A^ Skewed distribution; thus, Spearman’s correlation was calculated; for normally distributed variables, Pearson correlation was calculated; * *p*-value < 0.05; ** *p*-value < 0.01. BMI = Body Mass Index; HOMA-IR = Homeostatic Model Assessment for Insulin Resistance; HDL-C = High Density Lipoprotein Cholesterol; LDL-C = Low Density Lipoprotein Cholesterol; CRP = C-Reactive Protein; IL-6 = Interleukin-6; TNFα = Tumour Necrosis Factor-Alpha; ALT = Alanine-Aminotransferase.

## Data Availability

The data presented in this study are available upon request from the corresponding author.

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
