# Peer review of "Plasma Myostatin Increases with Age in Male Youth and Negatively Correlates with Vitamin D in Severe Pediatric Obesity"

_nutrients, 2022, doi:10.3390/nu14102133_

Round 1

Reviewer 1 Report

This study investigated whether myostatin and irisin are associated with metabolic parameters including the vitamin D status in pediatric patients with severe obesity. The authors found that myostatin concentrations, particularly in males, positively correlated with age and pubertal stage as well as metabolic parameters including insulin resistance. Irisin concentrations correlated positively with HDL and negatively with LDL cholesterol values. Basically, this study is interesting since it focused on pediatric patients with severe obesity. However, many studies have found that myostatin is increased in humans with obesity. Thus, the novelty of this study is limited. The study design and statistical analysis are good. 

1.The most important finding of this study is that myostatin had a negative correlation with Vit D levels in serum. However, the measurement method of Vit D levels in serum should be described in detail. The 25-(OH)-D3 is the major form of Vit D existing in blood. What and how did the authors detect the level of Vit D in serum?

2.The authors concluded that supplementation of Vit D or other dietetic and pharmacological interventions preventing a rise in myostatin levels could be therapeutic or preventive options in pediatric patients with severe obesity. The conclusion is not proper. No evidence is available to support that supplementation of Vit D could prevent the rise of myostatin level in severe obesity. The negative correlation between Vit D and myostatin does not assure that Vit D supplementation could lower myostatin.  

Author Response

1.The most important finding of this study is that myostatin had a negative correlation with Vit D levels in serum. However, the measurement method of Vit D levels in serum should be described in detail. The 25-(OH)-Dis the major form of Vit D existing in blood. What and how did the authors detect the level of Vit D in serum?

Thank you for the comment and sorry for this omission. It´s total 25-hydroxy vitamin D using automated chemiluminescent LIAISON® 25 OH Vitamin D TOTAL immunoassay from DiaSorin. We added the respective details to the methods section (lines 90-91).

2.The authors concluded that supplementation of Vit D or other dietetic and pharmacological interventions preventing a rise in myostatin levels could be therapeutic or preventive options in pediatric patients with severe obesity. The conclusion is not proper. No evidence is available to support that supplementation of Vit D could prevent the rise of myostatin level in severe obesity. The negative correlation between Vit D and myostatin does not assure that Vit D supplementation could lower myostatin.  

Answer: Thank you for this critical comment. It is clear that we do not provide evidence that supplementation of Vit D prevents the rise of myostatin level in severe obesity and we never intended to claim so. However, we believe that our work may induce research into that interesting direction. We reworded the respective statements (abstract lines 26-28, discussion lines 184-189 and 215-218) to avoid any impression of unjustified conclusions.

Reviewer 2 Report

Baumgartner et al report a correlation analysis between 3 myokines and multiple metabolic parameters in 108 severely obese children. They showed that myostatin was positively associated with age and puberty status as well as insulin resistance in male patients and inversely correlated to vitamin D levels, especially in female patients. Consistent reversed correlation were found with follistatin (an inhibitor of myostatin). Irisin correlated positively to HDL cholesterol and negatively with LDL cholesterol. They concluded that decreasing myostatin could be a target for prevention or treatment of severe obesity in children. They proposed that supplementation   in vitamin D could be such a treatment.

Generally speaking, the report addresses important questions regarding the clinical relevance of myokine parameters in obese children. The design of the study is quite sound, the methods adequate and the results convincing.

There are however some points that need to be addressed:

Line 78: do not start a sentence with number (e.g. 135)

Table 1, 2 and 3: use SI units

Line 127 and table 2 provide conflicting results for the correlation of myostatin with parathyroid hormone (negative in text and postivie in table).

Line 131: read Figure 1 (and not Figure 1A)

Line 131: show relts in figure 1 for follistatin as well.

Table 3 is not clear at all: please provide the meaning of the 3 results per cytokine (the guess was total sample adjusted with age and Tanner stage; in females, in males?)

Line 164: read “dysfunctions”

Line 175-190: this paragraph discussing the reversed correlation between vitamin d and myostatin should be improved. Correlation is not causation and the references used were not very convincing at linking both parameters. The authors should be more cautious at implying that supplementation in vitamin d could target a decrease in myostatin. They could suggest that this is a hypothesis that need to be tested with additional interventional studies (as stated in conclusion). In this line, the last sentence of the abstract should be rephrased accordingly.

Line 203: read “larger”

Author Response

There are however some points that need to be addressed:

  1. Line 78: do not start a sentence with number (e.g. 135)

Answer: Thanks for the hint. We reworded this sentence (line 79)

  1. Table 1, 2 and 3: use SI units

Answer: All data are now given in  SI units. Thank you!

  1. Line 127 and table 2 provide conflicting results for the correlation of myostatin with parathyroid hormone (negative in text and postivie in table).

Answer: Thank you for uncovering this conflict. We re-checked the data. There is a positive correlation between myostatin and parathyroid hormone, as now stated in the text (line 128)

  1. Line 131: read Figure 1 (and not Figure 1A)

Answer: Thank you for this comment, we corrected this.

  1. Line 131: show relts in figure 1 for follistatin as well.

Answer: The figure now also shows the follistatin results. Thank you for this suggestion.

  1. Table 3 is not clear at all: please provide the meaning of the 3 results per cytokine (the guess was total sample adjusted with age and Tanner stage; in females, in males?)

Answer: Thank you, that must have been lost in the transfer of the table. The subdivision into "all" "female" and "male" was added.

  1. Line 164: read “dysfunctions”

Answer: Thank you for this comment. The changes were made. (Now line 168)

  1. Line 175-190: this paragraph discussing the reversed correlation between vitamin d and myostatin should be improved. Correlation is not causation and the references used were not very convincing at linking both parameters. The authors should be more cautious at implying that supplementation in vitamin d could target a decrease in myostatin. They could suggest that this is a hypothesis that need to be tested with additional interventional studies (as stated in conclusion). In this line, the last sentence of the abstract should be rephrased accordingly.

Answer: Thank you for this important comment. We rephrased the abstract in line with your suggestion to avoid any impression of unjustified interpretation of our results.

  1. Line 203: read “larger”

Answer: We corrected this, thank you.